# Silicon–Gold Nanoparticles Affect Wharton’s Jelly Phenotype and Secretome during Tri-Lineage Differentiation

**DOI:** 10.3390/ijms23042134

**Published:** 2022-02-15

**Authors:** Elena V. Svirshchevskaya, Nina V. Sharonova, Rimma A. Poltavtseva, Mariya V. Konovalova, Anton E. Efimov, Anton A. Popov, Svetlana V. Sizova, Daria O. Solovyeva, Ivan V. Bogdanov, Vladimir A. Oleinikov

**Affiliations:** 1Shemyakin-Ovchinnikov Institute of Bioorganic Chemistry RAS, 16/10 Miklukho-Maklaya Str., 117997 Moscow, Russia; ninavsharonova@yandex.ru (N.V.S.); mariya.v.konovalova@gmail.com (M.V.K.); sv.sizova@gmail.com (S.V.S.); d.solovieva@mail.ru (D.O.S.); contraton@mail.ru (I.V.B.); voleinik@mail.ru (V.A.O.); 2National Medical Research Center of Obstetrics, Gynecology and Perinatology Named after Academician V. I. Kulakov of the Ministry of Health of the Russian Federation, 4 Oparina Str., 117997 Moscow, Russia; rimpol@mail.ru; 3Shumakov National Medical Research Center of Transplantology and Artificial Organs, 1 Schukinskaya Str., 123182 Moscow, Russia; esvir@yandex.ru; 4Institute of Engineering Physics for Biomedicine, National Research Nuclear University MEPhI, 31 Kashirskoe Sh., 115409 Moscow, Russia; aapopov1@mephi.ru

**Keywords:** silicon–gold nanoparticles, Wharton’s jelly mesenchymal stem cells, MSC, osteogenic, chondrogenic, adipogenic differentiation, cell phenotype, soluble factors, Luminex, G-CSG, VEGF

## Abstract

Multiple studies have demonstrated that various nanoparticles (NPs) stimulate osteogenic differentiation of mesenchymal stem cells (MSCs) and inhibit adipogenic ones. The mechanisms of these effects are not determined. The aim of this paper was to estimate Wharton’s Jelly MSCs phenotype and humoral factor production during tri-lineage differentiation per se and in the presence of silicon–gold NPs. Silicon (SiNPs), gold (AuNPs), and 10% Au-doped Si nanoparticles (SiAuNPs) were synthesized by laser ablation, characterized, and studied in MSC cultures before and during differentiation. Humoral factor production (*n* = 41) was analyzed by Luminex technology. NPs were nontoxic, did not induce ROS production, and stimulated G-CSF, GM-CSF, VEGF, CXCL1 (GRO) production in four day MSC cultures. During MSC differentiation, all NPs stimulated CD13 and CD90 expression in osteogenic cultures. MSC differentiation resulted in a decrease in multiple humoral factor production to day 14 of incubation. NPs did not significantly affect the production in chondrogenic cultures and stimulated it in both osteogenic and adipogenic ones. The major difference in the protein production between osteogenic and adipogenic MSC cultures in the presence of NPs was VEGF level, which was unaffected in osteogenic cells and 4–9 times increased in adipogenic ones. The effects of NPs decreased in a row AuNPs > SiAuNPs > SiNPs. Taken collectively, high expression of CD13 and CD90 by MSCs and critical level of VEGF production can, at least, partially explain the stimulatory effect of NPs on MSC osteogenic differentiation.

## 1. Introduction

Recent advances in nanotherapeutics provide diverse groups of synthetic nanoparticles (NPs) and other types of nanomaterials. Intravenous injection of NPs results in their accumulation in the liver [1,2,3], damaging epithelial and Kupfer’s cells. Liver as well as other organ regeneration depends on the functional potential of tissue stem cells. Mesenchymal stem cells (MSCs), which can be obtained from different tissues, are used as in vitro mimics of stem cells. MSCs are characterized by a set of specific surface marker, the ability to adhere to plastic, and to differentiate into different lineages depending on the factors. Multiple papers have demonstrated that different NPs affect the MSCs’ ability to differentiate into osteogetic and adipogenic lineages. Most consistent results show a stimulation of the osteogenic differentiation [4,5,6,7] and a suppression of the adipogenic one [7,8]. The chondrogenic differentiation of MSCs is likely to be unaffected by NPs as direct data are scarce [9].

The osteogenic differentiation of MSCs was enhanced by multiple nanosized material such as iron, silicon, gold, silver NPs, chitosan-collagen nanofibers, and by many other nanocomposites, showing that this effect is likely to be rather unspecific [10,11,12]. In spite of a sufficient amount of data being accumulated, the mechanisms of NP mediated effects on MSC differentiation have been poorly studied. There are several papers showing that autophagy, synthesis of chemokines, or induction of pro-inflammatory responses can be involved [13,14,15,16,17]. Limited data are available on pro- or anti-inflammatory properties of NPs in MSC cultures. As indirectly shown, NPs are more likely to induce anti-inflammatory responses [16,17].

Silicon is a part of various body tissues and is mostly important for bones, cartilage, and connective tissue well-being with a total amount of 7 g in the human body. Currently, a variety of SiNPs is developed for different biological applications [18,19,20]. Biodegradability and biocompatibility as well as the possibility to obtain SiNPs of various structures make silicon a promising material for use in the various fields of science, technology, and medicine. In medicine, SiNPs and its composites with organic and inorganic components have mostly been studied to deliver drugs to tumors [18,21] or to regenerate bone tissue [22]. Gold nanoparticles (AuNPs) are used in bionanotechnology due to the possibility of surface modification through a variety of functional groups. However, high concentrations of gold can be toxic due to its high affinity to sulfhydryl (SH) groups in SH-containing proteins [23]. Lee et al. demonstrated that AuNPs interfered with the formation of cytoskeletal structure, cell migration, inhibited DNA replication, and caused DNA damage via oxidative stress [24]. SiNPs are less toxic in, possibly due to a higher adsorption by tissues, however, at high concentrations, in vitro toxicity includes pro-inflammatory responses, oxidative stress, and autophagy [25]. Gold–silicon composites show acceptable toxicity and good biocompatibility [26].

The aim of this work was to study the effect of NPs on the functional characteristics of Wharton’s jelly MSCs during tri-lineage differentiation in the presence of NPs. For this study, we selected three types of NPs (SiNPs, AuNPs, and silica core and gold shell SiAuNPs) that were nontoxic in our previous experiments.

## 2. Results

### 2.1. Characterization of NPs

NPs were obtained by the method of pulsed laser ablation in liquid (PLAL) [27,28,29]. Earlier, we modified PLAL using solid silicon and gold targets [28]. The targets were rotated during PLAL in order to ensure homogenous ablation (Figure 1a). The concentration of Si and Au in NPs was estimated by the targets weighing before and after the ablation. Accuracy of this approach has been shown earlier [29]. SiNPs consisted of pure silicon (Figure 1b) while SiAuNPs contained 85–90% and 10–15% of Si and Au accordingly, depending of the batch of NPs (Figure 1c). Figure 1d shows a STEM image of a single SiAuNP where the large grey particle corresponds to SiNP with light spots of shelled gold on its surface. The diameter of SiNPs and SiAuNPs were 148 ± 37 and 185 ± 40 nm (Figure 1e,f). We were unable to measure the AuNP diameter, but their diameter was approximately 20–40 nm (Figure 1d). All NP types exhibited spherical shapes and smooth surfaces (Figure 1d).

### 2.2. Characterization of Mesenchymal Stem Cell Interactions with NPs

According to the International Society for Cellular Therapy, MSCs express a panel of characteristic clusters of differentiation (CD), namely CD73, CD90, CD105, and some others while they do not express immune cell associated markers CD14, CD19, CD34, major complex histocompatibility (MHC) class II HLA-DR, and many others [30]. At the same time, all MSCs express MHC class I HLA-ABC as a hallemark of all donor cells. Confocal microscopic analysis demonstrated a correct MSC profile (Figure 2a), which was also supported by flow cytometry analysis (Figure 3).

### 2.3. Cytotoxicity of NPs

Incubation of MSCs with the NPs resulted in the particle endocytosis. To visualize it, MSCs were incubated with NPs for 24 h, washed out from unbound NPs, and analyzed by spectrophotometry (Figure 2b). Higher optical densities of MSCs incubated with NPs demonstrated NP accumulation within the cells. Additionally, intracellular localization of NPs was detected by 2D scanning probe nanotomography [31] (Figure 2c).

Viability of MSCs was analyzed by the standard MTT assay, which demonstrated some toxicity of all three NP types at high doses (100 μg/mL) and a stimulatory effect at low ones (3–12 μg/mL) (Figure 2d). Possible NP toxicity can be a result of the production of reactive oxygen species (ROS). To analyze ROS production, a non-toxic dose of 25 μg/mL was selected based on the MTT data. Rather unexpectedly, all NPs at this concentration suppressed ROS production (Figure 2e).

Taken collectively, all studied parameters demonstrated the absence of toxic effects against MSCs in the range <25 μg/mL of Si and Au based NPs.

### 2.4. Direct Effect of NPs and Differentiation Factors on MSCs Phenotype

Differentiation of MSCs requires cell incubation with the specific stimuli for 21 days. During this long period, cells are not detached; the only treatment is the replacement of culture medium and differentiating factors. Analysis of the cell phenotype showed that the expression of the stem markers decreased significantly to day 21 while it was sufficiently preserved at day 14 (Appendix A). To analyze a direct effect of differentiation stimuli on MSC phenotype, the cells were seeded at low density without (control) and with chondrogenic, osteogenic, or adipogenic factors. The cells were incubated for 14 days, detached and used to measure MSC specific markers by flow cytometry. The only effect was detected in the chondrogenic cultures where CD73 and CD105 expression was decreased while the chondrogenic marker CD146 [32] increased (Figure 3a,b). During differentiation, HLA-ABC expression decreased in all stimulated cultures in a comparison with the untreated controls (Figure 3a).

The same approach was used to estimate the effect of NPs on MSC phenotype. The cells were incubated without (control) and with different NPs for 14 days with a regular medium exchange only, and NPs were only added at the beginning of the cultures. NPs per se did not affect the MSC markers significantly; Au-containing NPs increased CD73 expression, and all NPs decreased CD105 expression (Figure 3c,d).

### 2.5. Combined Effect of NPs and Differentiation Factors on MSC Phenotype

When MSCs were stimulated both with the differentiating factors and NPs, there were no changes in the expression of MSC markers CD73 and CD105 different from the undifferentiated control (data not shown). However, all NPs significantly increased the expression of CD90 and CD13 under osteogenic differentiation (Figure 3e,f) over the control cells incubated only with the differentiation stimuli (Dif. cont). CD13 antigen is a transmembrane protein detected on the surface of committed progenitor cells and is thus a marker of cell differentiation [33]. These results may reflect the enhancing effect of SiNPs or AuNPs on MSC osteogenic differentiation, as shown earlier

### 2.6. Humoral Factor Production by Wharton’s Jelly MSCs

Production of humoral factors may characterize the response to exogenous challenges such as NPs. A wide panel of humoral factors produced by MSCs under treatment with NPs, differentiation factors, or their combination was analyzed by Luminex technology. Analysis included 41 factors: nine growth factors, 11 chemokines, and 21 cytokines (Appendix A). Among all those analyzed, only 14 factors were detected (Figure 4, Figure 5, Figure 6 and Figure 7). Seven of them were produced in a range of 1600–10,000 pg/mL (Appendix A). These were G-CSF, VEGF (growth factors), GRO, MCP-1, MCP-3, IL-8 (chemokines), and IL-6 (cytokine). The second group, detected at 10–250 pg/mL, was represented by GM-CSF, PDGF-AA, FGF-2 (growth factors), fractalkine, RANTES, IP-10 (chemokines), and IL-4 (cytokine) (Appendix A). Only two cytokines were detected, IL-6 and IL-4, which both affect T and B lymphocyte differentiation and thus are paracrine factors.

Time dependent analysis of humoral factor production in control non differentiated cultures demonstrated that there was no overall decrease in factor levels during 21 days of cultivation (Appendix A).

### 2.7. Humoral Factor Production by Wharton’s Jelly MSCs during Differentiation

Tri-lineage differentiation of MSCs resulted in different patterns of humoral factor production (Figure 4a,b) and a significant (*p* < 0.01) decrease in total quantity of protein secreted in comparison with undifferentiated cells (Appendix A). The most pronounced decrease was found in adipogenic cultures followed by chondrogenic and osteogenic ones.

### 2.8. Humoral Factor Production by Wharton’s Jelly MSCs Induced by NPs

To analyze a direct effect of NPs on MSC secretome, cells were incubated with NPs for 14 days with regular half-volume changes of culture medium. NPs were added once at the start of the cultures at a non-toxic concentration of 25 μg/mL. The major effect of NPs was found in the enhancement of G-CSF, VEGF, GRO, and GM-CSF production when compared with the control supernatants (Figure 4c,d). In all cases, AuNP effects were the strongest followed by SiAuNPs and SiNPs.

### 2.9. Effect of NPs on Chondrogenic Differentiation

Supernatants from MSCs cultivated with different NPs and chondrogenic stimuli for 14 days were analyzed as described. In all cases, supernatants from the non-differentiated (shown in Figure 3a,b) and the differentiated without NPs cells were analyzed.

It appeared that SiNPs did not affect the chondrogenic or osteogenic cultures in a comparison with the differentiated controls (Appendix A). For clarity, only data for AuNPs and SiAuNPs are shown in chondrogenic and osteogenic cultures.

Both AuNPs and SiAuNPs stimulated G-CSF, GM-CSF, PDGF-AA, and FGF-2 growth factor production during chondrogenic differentiation (Figure 5a,b). An increase in IL-4 and MCP-3 concentrations was also observed. The major factors affected by NPs were G-CSF, GM-CSF, FGF-2 (>2 times), and PDGF-AA (>11 times).

To verify chondrogenic differentiation, MSCs were stained with Safranin O at day 21 of incubation. Safranin O stains cartilage in red (Figure 5c). We did not compare the level of Safranin O staining in different cultures. There was a high variability between wells, which can depend on the microcomposition of MSCs. Long time incubation of an initially low number of MSCs per well may heavily affect each culture.

### 2.10. Effect of NPs on Osteogenic Differentiation

Different effects of Au-containing NPs were registered in osteogenic cultures. Among group I factors, NPs slightly stimulated G-CSF production (>1.3 times) and around twice decreased VEGF in a comparison with the supernatants from the differentiated MSCs (Figure 6a). The major activity was found for AuNPs, which stimulated a panel of chemokines Fraktalkine, MCP-3, RANTES, and IP-10 (>1.7 times) (Figure 6b). SiAuNPs demonstrated minimal influence.

Osteogenic differentiation of MSCs was verified in the same way as above at day 21 using Alizarin Red staining (Figure 6c). Alizarin Red dye is commonly used to identify calcium containing osteocytes.

### 2.11. Effect of NPs on Adipogenic Differentiation

The most notable humoral effects of NPs were found in adipogenic cultures (Figure 7). All types of NPs including SiNPs affected humoral factor production more or less in the same manner. The most significant effect was an increase in VEGF and FGF-2 growth factors in the presence of Au containing NPs. At the same time, all NPs stimulated GM-CSF and a panel of chemokines (Figure 7a,b). All data were compared with the activity in the supernatants of the differentiated without NP cells.

Adipogenic differentiation of MSCs was also verified in the same way as above at day 21 using Oil Red O staining (Figure 7c). Oil Red O is used for staining drops of lipids in red color.

### 2.12. Comparison of NPs Effects on Tri-Lineage Differentiation of MSCs

To visualize and better understand the results obtained, we calculated the indexes of stimulation/inhibition of protein production when compared to the undifferentiated and the differentiated controls. Undifferentiated control was used to estimate the effect of differentiation on MSC production (Figure 8a). Differentiated controls were used to determine the effects of NPs (Figure 8b–d). Only those factors whose levels increased or decreased more than twice were selected. Such “responsive” to differentiation and/or NP proteins were G-CSF, VEGF, MCP-3 (group I) and GM-CSF, RANTES, and IP-10 (group II). Differentiating stimuli severely inhibited the production of all of them but VEGF, which was unaffected in chondrogenic and osteogenic cultures (Figure 8a). NPs mostly demonstrated a suppressive effect in the chondrogenic cultures (Figure 8b) while they stimulated the osteogenic and the adipogenic ones (Figure 8c,d accordingly). AuNPs demonstrated the highest effect followed by SiAuNPs. The only evident difference between the osteogenic and the adipogenic cultures treated with NPs was in the VEGF levels, which significantly increased during adipogenesis and were unchanged in osteogenic cultures. Among the selected proteins, total concentrations of G-CSF and VEGF in all cultures were more than 10 times higher than of all the others. We hypothesize that VEGF and G-CSF are major players in the NPs and MSC interactions during the differentiation.

## 3. Discussion

Multiple studies have shown that NPs and other particulate materials are able to stimulate osteogenic differentiation of MSCs, inhibit adipogenesis, and are likely not to affect chondrogenesis [4,5,6,7,8,9]. The mechanisms of these effects are modernly obscure, while their better understanding may help to improve chondrocyte and osteocyte generation for clinical applications [34,35,36].

Incubation of MSCs with both differentiation stimuli and NPs resulted in a significant increase in CD13/CD90 expression observed only under osteogenic differentiation, which correlates with the earlier found enhancing effect of NPs on osteocytes. CD13 is a membrane bound type II metalloprotease, expressed by many cells including MSCs [33], participating in cell differentiation. Sorted murine CD90 (Thy1+) cells also demonstrated an enhanced osteogenic differentiation [37]. There was no selectivity between NPs as all three NP types demonstrated the same activity. The role of CD13/CD90 increased expression by MSCs during osteogenic differentiation in the presence of NPs was not shown earlier and requires further investigation.

Humoral factors play major roles in tissue organization and cell-to-cell cross talk. Therefore, we tried to estimate the role of humoral factors produced by MSCs during their differentiation in the presence of NPs.

Our analysis identified two groups of humoral factors constantly produced at significantly different levels by MSCs in both steady-state cultures and during differentiation (Appendix A). Group I factors can be considered as homeostatic ones controlling cell functions such as proliferation (G-CSF), angiogenic signaling (VEGF), niche formation (GRO, IL-8, MCP-1), cell senescence (IL-6) [38], while group II factors are likely to serve as a reaction to the local needs or challenges. Data on MSCs secretome are limited and heterogeneous [39,40,41,42]. Increased production of G-CSF, GM-CSF, IP-10, MIP-1α, and MCP-3 during osteogenesis was shown by several groups [39,40]. However, concentrations of different factors vary significantly from paper to paper [41,42], possibly due to the different commercial kits used.

Major effects found by us are summarized in Figure 8. During differentiation of MSCs, significant decrease was found in major growth factors G-CSF, VEGF, GM-CSF, and chemokine MCP-3, RANTES, and IP-10 levels. Co-cultivation of MSCs with NPs during cell differentiation demonstrated three different patterns of humoral factor production. For the exception of a low stimulatory effect on G-CSF production, a suppressive effect was found in chondrogenic cultures. In contrast, AuNPs stimulated protein production in MSC osteogenic and adipogenic cultures. The major difference between osteogenesis and adipogenesis was found in the VEGF level, which was unaffected in osteogenic and increased in adipogenic cultures. We hypothesize that these three types of NP effects on humoral factor production are related to the different influence on MSC differentiation, as shown earlier. Possibly a balance between VEGF and G-CSF is the most important regulator of MSC differentiation affecting both osteogenic and adipogenic differentiation. An increase in VEGF production by NPs in adipogenic cultures possibly blocks the differentiation, a phenomenon observed earlier.

In this work, we used three types of NPs. Among them, AuNPs demonstrated the highest effect on MSCs secretome, while SiNPs only induced an increased CD13/CD90 expression by MSCs. At the same time, a stimulatory effect of silica NPs, mesoporous silica NP-based films, and other silica nanocomposites on osteogenic differentiation of MSCs has been shown by multiple groups [5,6,43,44,45]. Consequently, an increased CD13/CD90 expression induced by SiNPs can be responsible for osteogenesis stimulatory effects. The question of whether the increased CD13/CD90 expression or the effects on MSC secretome is more important in the pro-osteogenic effect of NPs should be resolved in further studies.

## 4. Methods

### 4.1. Synthesis of SiNPs

Pure silicon nanoparticles (SiNPs) were synthesized by ultra-shot (fs) laser ablation in water as described by Riedel et al. [27] and modified by us [28,29]. Scheme of the ablation geometry is drawn in Figure 1(a1). For the synthesis of SiNP, a piece of a single crystal Si wafer (Telecom-CTB, Moscow, Russia, N-doped, 1–10 Ω cm) was fixed in a glass (BK-7) vessel filled with 14 mL of ultrapure water (18.2 MΩ * cm at 25 °C). A 3 mm diameter beam from a Yb:KGW laser (1030 nm wavelength, 270 fs pulse duration, up to 400 µJ pulse energy, 1–100 kHz, TETA 10 model, Avesta, Moscow, Russia) was focused by a 75 mm lens on the surface of the target. The thickness of the liquid layer was kept at 10 mm above the surface of the target. Position of the focusing lens was adjusted to obtain maximum productivity of the ablation process, as measured by weighting the target before and after the ablation. Duration of each experiment was 30 min. The ablation vessel was mounted on a platform that performed a continuous scanning over a 2 mm × 2 mm area with a 5 mm/s speed using motorized linear translational stages (Newport, Every, France) in order to avoid ablation from the same area.

### 4.2. Synthesis of AuNPs

AuNPs were synthesized using the same laser in the ablation geometry as described above (Figure 1(a2)). For this, a gold target (99.99%, Sigma Aldrich, Saint-Quentin Fallavier, France) immersed in 7 mL of aqueous 0.1% NaCl solution was irradiated by a focused laser beam. To ensure a homogenous ablation process, the ablation target was continuously moved with 2 mm/s speed using motorized linear translational stages (Newport, Every, France).

### 4.3. Synthesis of SiAuNPs

To develop core-shell SiAuNPs, the suspension of SiNPs obtained at step 1 (Figure 1(a1)) was placed in the glass vessel over the gold target and ultra-shot laser ablation was fulfilled (Figure 1(a3)).

### 4.4. Characterization of Nanoparticles

Morphology, structure, and size of synthesized SiNPs were characterized by a scanning transmission electron microscopy (STEM) system (TESCAN MAIA 3, Brno, Czech Republic) operated at 0.1–30 kV. Samples for electron microscopy were prepared by dropping 1 μL of a NP solution onto a cleaned silicon substrate and subsequently dried at ambient conditions. Concentrations of NP solutions were determined by measuring target weight before and after the ablation step and dividing this mass difference on the ablation liquid volume.

X-ray spectral data were obtained using a scanning electron microscope TESCAN MIRA 3 (Brno, Czech Republic) at electron energy of 20 eV.

Raman spectrum of NPs was recorded on a Renishaw instrument using a QONTOR inVia microscope (Woodview, UK). Drops of colloidal NPs (1 mg/mL) were applied to the quartz surface. The spectrum was recorded after drying at the excitation wavelength of 532 nm, and the power on the sample was 1 MW. The peak at 520 nm is characteristic of crystalline silicon.

The diameters of NPs were determined by dynamic light scattering (90 Plus Particle Size Analyzer Brookhaven Instruments Corporation, Vernon Hills, IL, USA). All measurements were performed using 661 nm laser light at room temperature with 90° angle of detection.

### 4.5. Wharton’s Jelly MSC Production and Characterization

The primary culture of MSCs was isolated from the Wharton’s jelly of the umbilical cord tissue. The material was collected with the written informed consent of healthy examined women in labor or after cesarean section. Experiment protocol was approved by the Institutional Ethical Commission, #233 from 02.12.2020. Isolation of MSCs was carried out in accordance with the protocol described earlier [46]. The cells were cultured in an atmosphere of 5% CO_2_ at 37 °C in DMEM/F12 medium (Gibco, Waltham, MA, USA) with the addition of 10% fetal calf serum, 2 mM L-glutamine, 100 U/mL of penicillin, and 100 μg/mL of streptomycin (PanEco, Moscow, Russia) in 25 cm^2^ culture flasks (Corning, NY, USA). Culture medium was replaced by 50% every three days. When 80% confluence was reached, the cells were detached using trypsin/EDTA (PanEco, Moscow, Russia) and split at a ratio of 1 to 2. The expression of MSC surface markers was analyzed by flow cytometry of confocal microscopy using primary antibodies conjugated with phycoerythrin (PE) to CD13, CD19, CD73, CD90, CD105, CD146, HLA-DR, and conjugated with FITC HLA-ABC (BD, USA, Franklin Lakes, NJ, USA), according to the manufacturer’s protocol.

### 4.6. Confocal Fluorescence Imaging

MSCs (10^3^) were grown in 200 μL drop on sterile cover slides placed in 6-well plates (Costar, Washington, WA, USA). After adhesion of the cells overnight, the cells were incubated with specific antibodies for 1 h at 5% CO_2_ at 37 °C. The cells were then washed, fixed with 4% paraformaldehyde (PFA) for about 15 min, washed again, and polymerized with Mowiol 4.88 medium (Calbiochem, Nottingham, UK). Hoecst 33342 (Merk, Darmstadt, Germany) was used to visualize the nuclei. Slides were analyzed using Eclipse TE2000 confocal microscope (Nikon, Tokyo, Japan).

### 4.7. Flow Cytometry

For the analysis of MSC surface markers, cells were detached as described, transferred (2 × 10^3^ cells per well) into round bottom 96-well plates in PBS, 0.05% NaN_3_, and 1% bovine albumin (FACS buffer), incubated with specific antibodies for 1 h at +4 °C, washed, and measured on a FACSCalibur device (BD, Bergen, NJ, USA). The cytometric data were analyzed by WinMDI2.8 software.

### 4.8. Scanning Probe Nanotomography (SPNT)

To visualize NPs inside the cells, MSCs were loaded with 25 μg/mL NPs for 24, fixed with 4% PFA, and analyzed by the SPNT method as described [36] using a Reichert–Jung Ultracut E ultramicrotome (Leica Microsystems GmbH, Vienna, Austria) adapted for the use of a specially designed SPM head.

Spectrophotometry was used to estimate NP endocytosis by MSCs. To this end, cells were incubated with 25 μg/mL NPs overnight, washed three times with saline, plated on 96-well plates (Costar, Washington, WA, USA), and incubated overnight. Optical density (OD) was measured at 620 nm (MultiScan FC, ThermoScientific, Waltham, MA, USA).

### 4.9. In Vitro Cytotoxicity/Viability

Cytotoxicity/viability of NPs was investigated by the MTT assay [47]. Si-, Au-, and SiAuNPs dispersions were titrated on 96-well flat-bottom culture grade plates to yield the concentrations from 100 to 0.2 µg/mL. In prior experiments, NPs were treated by ultrasound to obtain homogenous dispersions. The cells were seeded at 3000 cell/well and incubated for 72 h in CO_2_-incubator at 37 °C. MTT was added to each well for the last 3 h. Culture medium was eliminated from the wells, and formazan crystals were dissolved in 100 μL of DMSO for 20 min. The optical densities were measured at 540 nm using a plate reader MultiScan FC (ThermoScientific, Waltham, MA, USA), and inhibition indices (II) were calculated as follows II = 1 − OD_exp_/OD_cont_, where OD_exp_ and OD_cont_ are the optical densities of the experimental and control wells respectively. The experiments were repeated several times and average data are shown.

### 4.10. Reactive Oxygen Species (ROS) Production

MSCs were seeded in 96-well flat-bottom plates at 3000 cells/well and incubated in aa CO_2_-incubator at 37 °C; overnight to reach an adhesive state. NPs (25 µg/mL) were added for 2 and 4 h. The ROS probe 2′,7′-dichlorodihydrofluorescein diacetate (DCF, Merk, Darmstadt, Germany) was added simultaneously with NPs. Analysis of fluorescence was conducted by a GlomaxMulti spectroluminometer (Promega, Fitchburg, MA, USA) at 488 nm. The results are shown as OD minus background without DCF.

### 4.11. Multiplex Analysis of Cytokines

The standard 41-plex human cytokine/chemokine magnetic bead panel using FLEXMAP 3D cytometer (EMD Milipore, Billerica, MA, USA). To this end, supernatants from MSCs were collected and analyzed according to the manufacturer’s instructions. Data were analyzed automatically with xPONENT software (EMD Millipore, Billerica, MA, USA). The list of cytokines is given in Appendix A.

### 4.12. MSCs Differentiation

MSCs were stimulated to differentiate into chondrocytes, osteocytes, and adipocytes using a StemPro Kit (Gibco, ThermoScientific, Waltham, MA, USA USA) according to the manufacturer’s instructions. In brief, MSCs from the second or third passages were seeded in 24-well plates (Costar, Duarte, WA, USA) at 10^4^ per well (three replicas per point) in DMEM/F12 10% fetal calf serum, 2 mM L-glutamine, 100 U/mL of penicillin, and 100 μg/mL of streptomycin (PanEco, Moscow, Russia) overnight for adhesion. Medium was then replaced for StemPro and differentiating factors were added. NPs were added at 25 μg/mL. Medium and differentiating factors were replaced 50% twice a week. Additional NPs were not added. Supernatants were collected at days 4, 10, 14, and 21 of incubation and frozen for further humoral factor analysis. Staining for osteogenic, chondrogenic, and adipogenic differentiation was fulfilled at day 21 of incubation using Safranin O, Alizarin Red, or Oil Red (all from Merk, Darmstadt, Germany) using standard protocols.

### 4.13. Statistics

Graphs were created using MS Excel software. The data are represented as mean ± SEM of at least two independent experiments or as one representative experiment from two. Statistical analysis was performed using the Student’s *t*-test. Significance levels of *p* < 0.05 were considered statistically reliable.

## Figures and Tables

**Figure 1 ijms-23-02134-f001:**
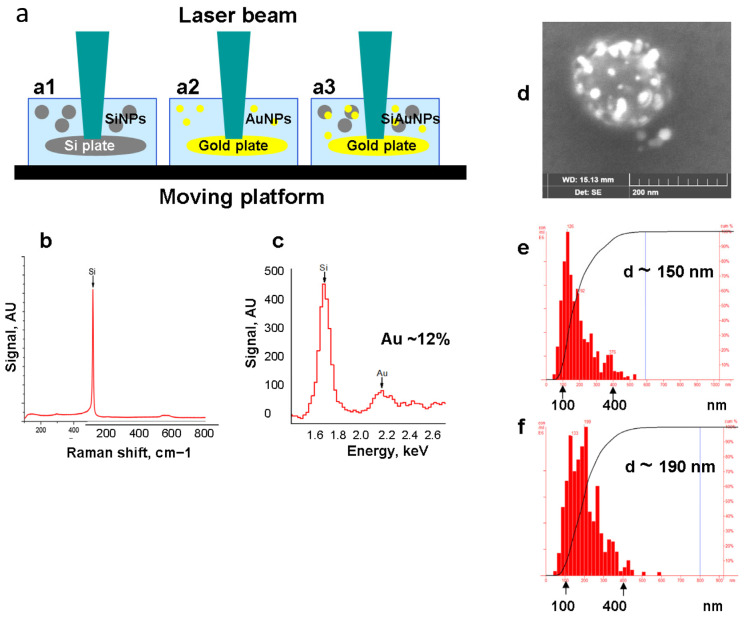
Synthesis and characterization of nanoparticles. (**a**) Scheme of laser ablation synthesizing SiNPs (**a1**), AuNPs (**a2**), and SiAuNPs (**a3**). (**b**,**c**) Chemical composition of SiNPs (**b**) and SiAuNPs (**c**). (**d**) STEM image of a SiAuNP where AuNPs locate on the surface of SiNP core. (**e**,**f**) Dynamic light scattering of SiNPs (**e**) and SiAuNPs (**f**).

**Figure 2 ijms-23-02134-f002:**
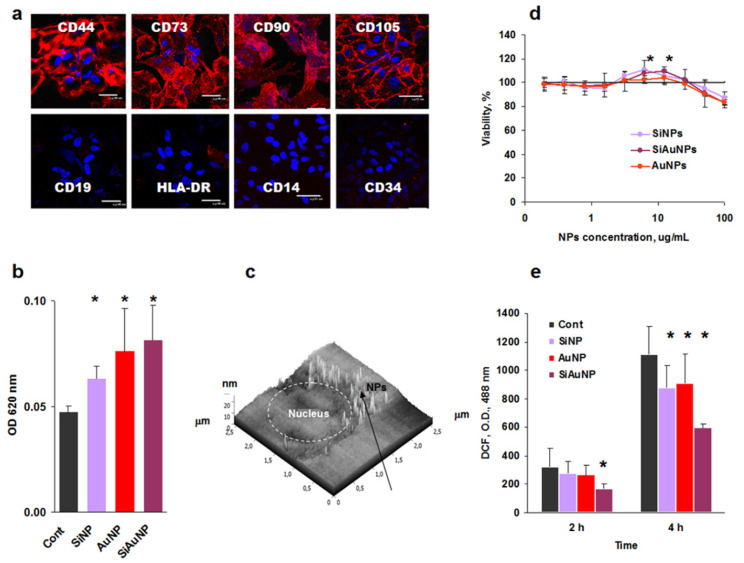
Characterization of mesenchymal stem cell interactions with NPs. (**a**) Phenotype of Wharton’s jelly MSCs. (**b**) Optical density (OD) of MSCs incubated with NPs for 24 h. (**c**) A representative image of SiAuNPs intracellular localization (arrow). (**d**) Effect of NPs on MSCs proliferation. (**e**) ROS production by MSCs in the presence of 25 μg/mL of SiAuNPs. Significant differences between the effects of each NPs vs. untreated controls are shown with asterisks.

**Figure 3 ijms-23-02134-f003:**
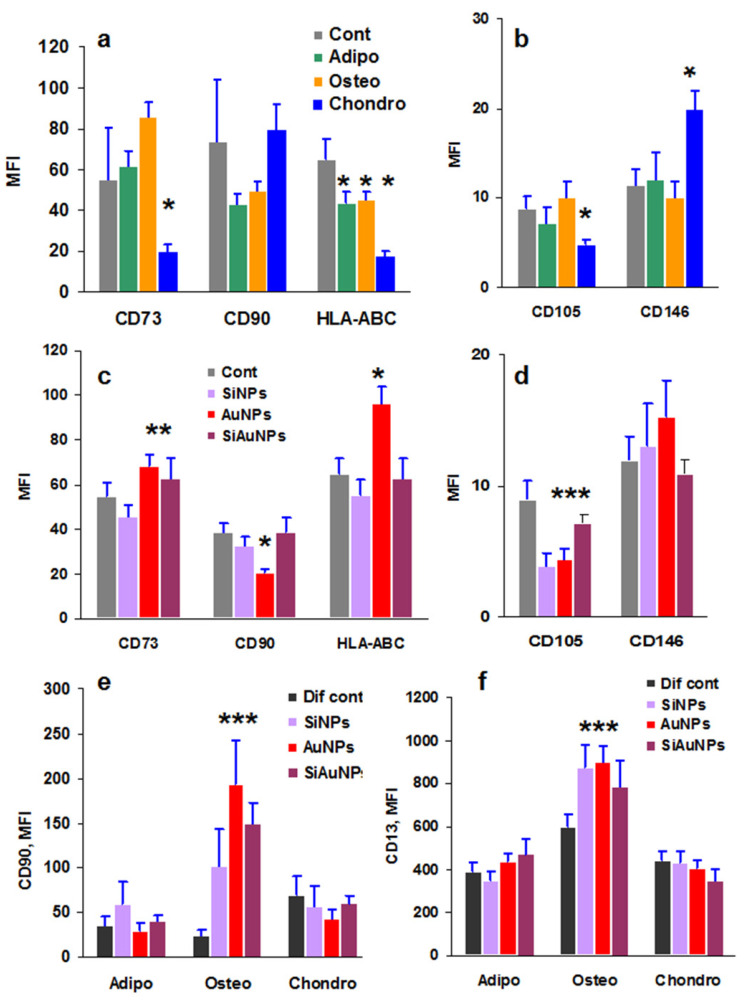
Surface markers of MSCs treated with NPs or/and differentiation factors for 14 days. (**a**,**b**) Expression of MSCs specific markers during 14 days of differentiation induced by specific adipogenic (Adipo), osteogenic (Osteo), or chondrogenic (Chondro) factors. (**c**,**d**) Effects of 25 μg/mL SiNPs, AuNPs, or SiAuNPs on MSC phenotype. (**e**,**f**) Expression of CD90 (**e**) and CD13 (**f**) by MSCs during differentiation in the presence of 25 μg/mL SiNPs, AuNPs, and SiAuNPs. Significant differences of each NPs vs. differentiation control (Dif Cont) (<0.05) are shown with asterisks. MFI designates mean fluorescence intensity. Data are shown from a representative experiment of two.

**Figure 4 ijms-23-02134-f004:**
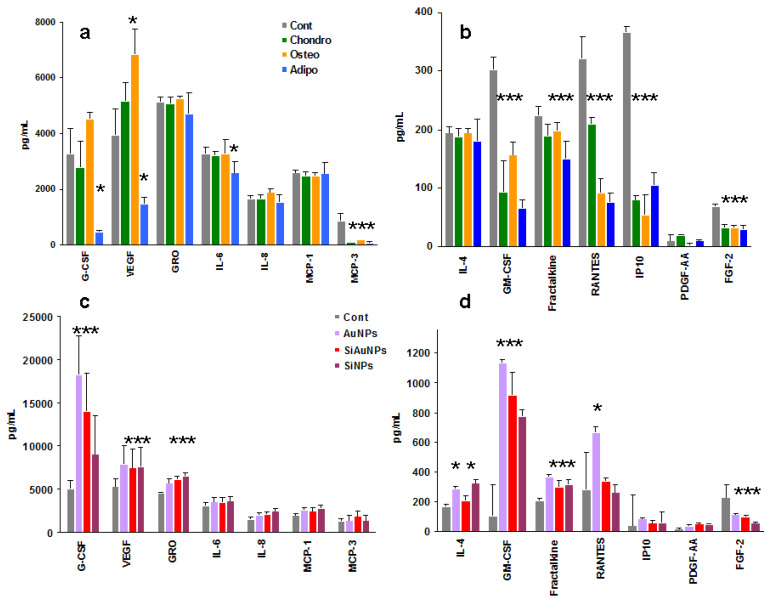
Effect of differentiation stimuli and NPs on the production of humoral factors by MSCs. Effect of chondrogenic (Chondro), osteogenic (Osteo), and adipogenic (Adipo) differentiation (**a**,**b**) or NPs (**c**,**d**) on group I (**a**,**c**) and II (**b**,**d**) factor production by MSCs at day 14 of cultivation. Significant differences in the values of individual factors in the differentiated MSCs supernatants vs. controls (<0.05) are shown with asterisks. Data are pooled from two independent experiments.

**Figure 5 ijms-23-02134-f005:**
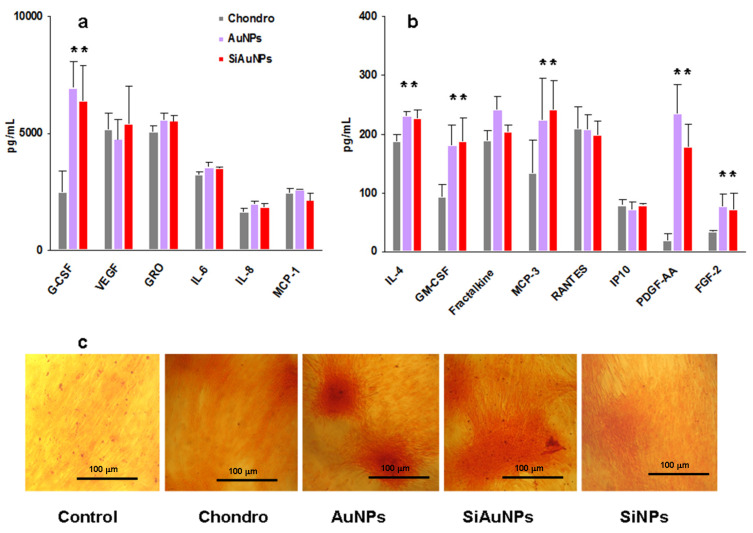
Humoral factors produced by MSCs during chondrogenic differentiation. (**a**,**b**) Group I (**a**) and II (**b**) factor production by MSCs on day 14 of differentiation in the presence of 25 μg/mL of AuNPs or SiAuNPs. Significant differences in the values of individual factors in the MSCs supernatants treated with NPs vs. differentiated controls (<0.05) are shown with asterisks. Data are pooled from two independent experiments. (**c**) Specific Safranin O staining of untreated (control) and differentiated cultures without (chondro) or with NPs at day 21. Scale bar 100 μm.

**Figure 6 ijms-23-02134-f006:**
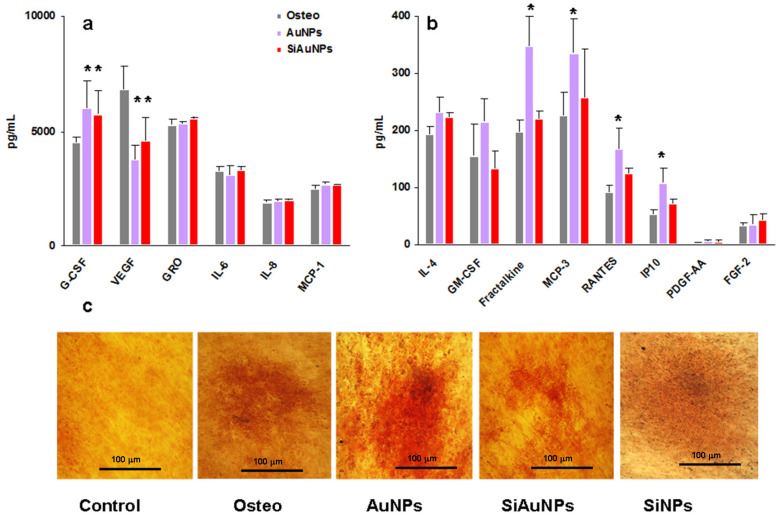
Humoral factors produced by MSCs during osteogenic differentiation. (**a**,**b**) Group I (**a**) and II (**b**) factor production by MSCs on day 14 of differentiation in the presence of 25 μg/mL of AuNPs or SiAuNPs. Significant differences in the values of individual factors in the MSCs supernatants treated with NPs vs. differentiated controls (<0.05) are shown with asterisks. Data are pooled from two independent experiments. (**c**) Specific Alizarin Red staining of untreated (control) and differentiated cultures without (osteo) or with NPs at day 21. Scale bar 100 μm.

**Figure 7 ijms-23-02134-f007:**
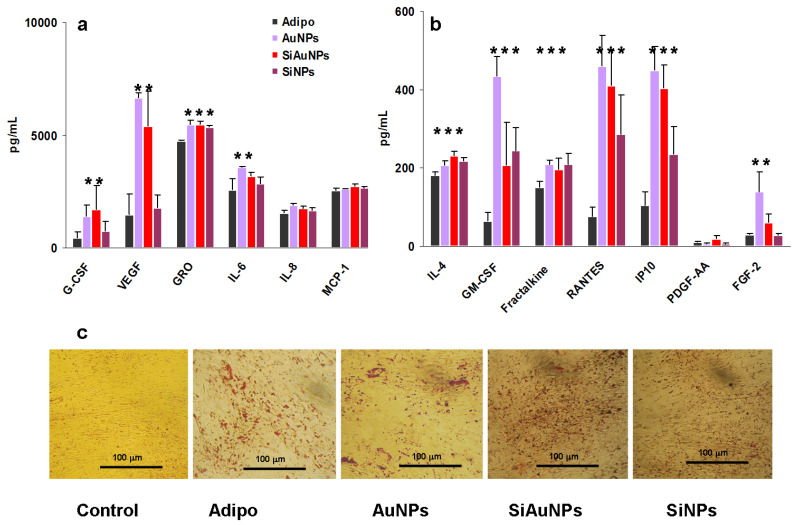
Humoral factors produced by MSCs during adipogenic differentiation. Group I (**a**) and II (**b**) factor production by MSCs on day 14 of differentiation in the presence of 25 μg/mL SiNPs, AuNPs, and SiAuNPs. Significant differences in the values of individual factors in the MSCs supernatants treated with NPs vs. differentiated controls (<0.05) are shown with asterisks. Data are pooled from two independent experiments. (**c**) Specific Oil Red O staining of untreated (control) and differentiated cultures without (adipo) or with NPs at day 21. Scale bar 100 μm.

**Figure 8 ijms-23-02134-f008:**
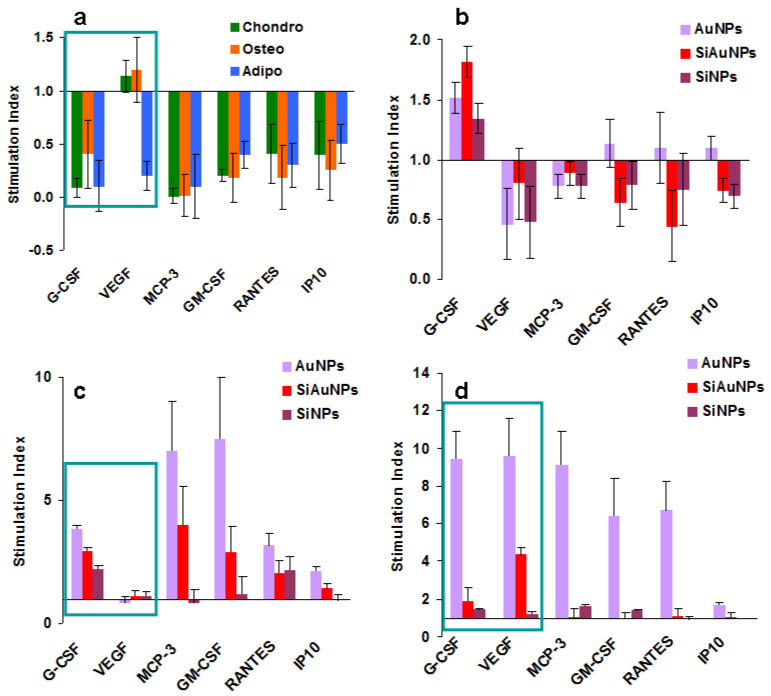
Significant changes in protein secretion by MSCs during differentiation. (**a**) Ratios of humoral factor levels in differentiated MSC cultures to control ones. (**b**–**d**) Effect of NPs on humoral factor production during chondrogenic (**b**), osteogenic (**c**), and adipogenic (**d**) differentiation in comparison with differentiating control cultures. The most important factors are framed.

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
