# Peer review of "Silicon–Gold Nanoparticles Affect Wharton’s Jelly Phenotype and Secretome during Tri-Lineage Differentiation"

_ijms, 2022, doi:10.3390/ijms23042134_

Round 1

Reviewer 1 Report

The authors have synthesized three kinds of nanoparticles (SiNPs, AuNPs and SiAuNPs) and assessed the effect of these NPs on MSC phenotype, humoral factor production by Wharton’s jelly MSCs, chondrogenic differentiation and so on. This study has achieved good efforts on NPs influences, however, there are many mistakes in the manuscript. So I recommend to you that major revision is needed before the manuscript is published. Corresponding comments are as follows:

  1. Figure 1b shows that the diameter of a single SiAuNPs is around 200 nm and the diameter of AuNPs is around 20 nm. While, it is better to evaluate the diameter of NPs using the average diameter of a large number of NPs rather than a single NPs.
  2. The authors demonstrate in line 126 that “The results for different types of NPs were the same (data not shown).”. Corresponding results should be added in the supplemental information.
  3. It is better to show the MTT results using survival rate as the y-axis as shown in Figure 2c.
  4. The information of Figure 2 is not complete and I cannot get full information in Figure 2b and 2e.
  5. The legend of Figure 3 shows that “a-b: Expression of MSC specific markers after incubation with 25 g/mL SiNPs, AuNPs, and SiAuNPs for 14 days.”. While there is not any information of NPs in Figure 3a-b, different colored of columns indicate Cont, Adipo, Oseto and Chondro. This is controversy. The authors need to explain.
  6. The authors has done a lot of experiments, but I still don’t know which NPs is better at specific function. The authors should add more information in the Conclusions part in order to make it easier for readers to understand.
  7. There are so many mistakes in the manuscript and professional editing needs to be performed.

Author Response

Thank you for the comments and found problems in our text. Please find below our responses.

Comments and Suggestions for Authors

The authors have synthesized three kinds of nanoparticles (SiNPs, AuNPs and SiAuNPs) and assessed the effect of these NPs on MSC phenotype, humoral factor production by Wharton’s jelly MSCs, chondrogenic differentiation and so on. This study has achieved good efforts on NPs influences, however, there are many mistakes in the manuscript. So I recommend to you that major revision is needed before the manuscript is published. Corresponding comments are as follows:

  1. Figure 1b shows that the diameter of a single SiAuNPs is around 200 nm and the diameter of AuNPs is around 20 nm. While, it is better to evaluate the diameter of NPs using the average diameter of a large number of NPs rather than a single NPs.

R1 We have added DLS data in Fig. 1, e, f.

  1. The authors demonstrate in line 126 that “The results for different types of NPs were the same (data not shown).”. Corresponding results should be added in the supplemental information.

R2 We have added data on NPs endocytosis in Fig. 2c.

  1. It is better to show the MTT results using survival rate as the y-axis as shown in Figure 2c.

We have reverted data in Fig. 2 d

  1. The information of Figure 2 is not complete and I cannot get full information in Figure 2b and 2e.

All legends were corrected

  1. The legend of Figure 3 shows that “a-b: Expression of MSC specific markers after incubation with 25 g/mL SiNPs, AuNPs, and SiAuNPs for 14 days.”. While there is not any information of NPs in Figure 3a-b, different colored of columns indicate Cont, Adipo, Oseto and Chondro. This is controversy. The authors need to explain.

Same

  1. The authors has done a lot of experiments, but I still don’t know which NPs is better at specific function. The authors should add more information in the Conclusions part in order to make it easier for readers to understand.

We ourselves are not sure which function is more specific. We have found two phenomena: i) increase in CD13 and CD90 during osteogenic differentiation in the presence of all NPs and ii) multiple humoral factor responses to AuNPs and SiAuNPs but not SiNPs. We have stated both possibility, more research is needed. 

  1. There are so many mistakes in the manuscript and professional editing needs to be performed.

We have edited the text.

Reviewer 2 Report

The manuscript reports interesting results about synthesizing three types of NPs (SiNPs, AuNPs, and silica 89 core and gold shell SiAuNPs) and studying their cytotoxicity, effect on phenotype, and hu- 90 moral factor production during three lineage differentiation of Wharton’s jelly MSCs.

This experiment is very interesting and useful, but the author has not properly explained the contribution of this study to the research field. Moreover, why the author choose Si and Au Nanomaterials? I encourage author to improve the paper for English language. Overall, the experimental design and results are not novel, bring new knowledge, and are not good enough for publication. However, I recommend substantial revision according to some specific comments on the manuscript below.

Title

It looks slightly longer and scientifically does not make sense. Please improve the title

Abstract

  1. Please briefly write down background of the study.
  2. Please highlight the novelty of your work
  3. Please clearly mention the significance of your study.
  4. Identify the future/ industrial application of this study.

Introduction

  • The introduction is weak. The authors started with why each of the stress they investigated was important and relevant, but I think a more convincing argument should be why those two NMs were selected?. 
  • The proper logic and importance of this study need improvement.
  • I also suggest you add another recent study regarding relevant literature (doi:10.3390/molecules24213916) here.
  • Many sentences are not understandable. Please overhaul the manuscript.
  • Authors are not aware of recent publications related to this research direction. Some latest literature might have been missed; please make sure to update your references. 
  • Please add more about selection of your NMs and their background as well. 1007/s11356-017-96652,10.1002/clen.201900228,  10.1016/j.envpol.2019.113032
  • Please revise your objective statement.
  • Please mention the novelty and hypothesis as well at the end of the introduction.

Results and discussion

  • What’s new finding in your synthesis methods? How your findings are novel and cost-effective as compared to previous reports.
  • Why did you choose 25 mg/mL exposure dose? Please address your selection criteria?
  • The discussion is less coherent and comprehensive, the authors need to look at the various aspect under different treatment conditions as a panorama and provide deeper discussion on the role of NMs effects.
  • 3 and fig 4 (C-d) need to change the color scheme as current graphs not visible and understandable.
  • Add a diagram illustrating a general mechanism behind your results.
  • During co-exposure of NMs treatment, have you performed any controlled experiment?
  • Please compare the toxic effects with other NMs and add the in-depth discussion. 1007/s11356-020-09565-8

Conclusion

  1. Conclusion is just a repetition of results. Please significantly improve this part.
  2. What’s practical application of your study? What’s new addition of your current experiment in terms of knowledge?
  3. Please add the mechanist illustration of your findings at the end.

Author Response

We are thankful to our reviewers for quick and significant comments. Please find below our responses.

Q1 Moreover, why the author choose Si and Au Nanomaterials?

We have used in our routine work many types of NPs including these ones, many were toxic. We have selected Si and Au NPs for this exact work only because we have a financial support to develop these types of NPs for further clinical application. These NPs are good, non toxic, easy to be produced, have good shelf-life. We have explained in the introduction their good quality determined earlier by many authors.  

Q2 I encourage author to improve the paper for English language.

I have polished the language.

Q3 Overall, the experimental design and results are not novel, bring new knowledge, and are not good enough for publication.

We have carefully screened the literature on any functional effects of NPs during MSCs differentiation and found only scattered. We are absolutely sure that presented data are new. 

  1. Title It looks slightly longer and scientifically does not make sense. Please improve the title

R1. The title (Mesenchymal stem cell secretome during differentiation in the presence of silicon-gold nanoparticles) was not long initially and contained sufficient information. Still we replaced it for “Silicon-gold nanoparticles affect Wharton’s jelly phenotype and secretome during tri-lineage differentiation”. The title describes what we have done.

  1. Abstract
    • Please briefly write down background of the study.

R2.1 We have revised the abstract completely.

Multiple studies demonstrated that various nanoparticles (NPs) stimulate osteogenic differentiation of mesenchymal stem cells (MSCs) and inhibit adipogenic one. The mechanisms of these effects are not determined. The aim of this paper was to estimate Wharton’s Jelly MSCs phenotype and humoral factor production during tri-lineage differentiation per se and in the presence of silicon-gold NPs. Silicon (SiNPs), gold (AuNPs) and 10% Au-doped Si nanoparticles (SiAuNPs) were synthesized by laser ablation, characterized, and studied in MSCs cultures before and during differentiation. Humoral factor production (n=41) was analyzed by Luminex technology. NPs were nontoxic, did not induce ROS production, and stimulated G-CSF, GM-CSF, VEGF, CXCL1 (GRO) production in 4 day MSCs cultures. During MSCs differentiation all NPs stimulated CD13 and CD90 expression in osteogenic cultures. MSCs differentiation resulted in a decrease in multiple humoral factor production to day 14th of incubation. NPs did not affect the production in chondrogenic cultures and stimulated it both in osteogenic and adipogenic ones. The major difference in the protein production between osteogenic and adipogenic MSCs cultures in the presence of NPs was VEGF levels which was unaffected in osteogenic cells and 4-9 times increased in adipogenic ones. The effects of NPs decreased in a row AuNPs>SiAuNPs>SiNPs. Taken collectively, high expression of CD13 and CD90 by MSCs and critical level of VEGF production can, at least, partially explain the stimulatory effect of NPs on osteogenic differentiation. 

2.2 Please highlight the novelty of your work

R2.2 We have commented in the text all major new points namely: i) an increase in CD13 and C90 expression by MSCs during osteogenic differentiation in the presence of all three types of NPs; ii) MSCs secretome was studied in dynamics of tri-lineage differentiation was done for the first time; iii) we have found the difference in the secretome between osteogenic and adipogenic differentiation; iv) we have shown the superior effect of AuNPs on the secretome. 

2.3. Please clearly mention the significance of your study.

R2.3 Our results give new directions for further studies

  • Identify the future/ industrial application of this study.

Osteocyte and chondrocyte in vitro production have clinical application.  

We have tried to include some of this information in the “Abstract”, however the word limit does not let us including all. Other points are highlighted in the paper body.

Introduction

 The introduction is weak. The authors started with why each of the stress they investigated was important and relevant, but I think a more convincing argument should be why those two NMs were selected?. 

We have revised “Introduction” pointing more attention to NPs effects on MSCs differentiation.

The proper logic and importance of this study need improvement.

We have done our best to improve it.

I also suggest you add another recent study regarding relevant literature (doi:10.3390/molecules24213916) here.

This paper is nether relevant to SiAuNPs nor to MSCs. We have included several new relevant studies.

Ahmad MA, Yuesuo Y, Ao Q, Adeel M, Hui ZY, Javed R. Appraisal of Comparative Therapeutic Potential of Undoped and Nitrogen-Doped Titanium Dioxide Nanoparticles. Molecules. 2019 Oct 30;24(21):3916. doi: 10.3390/molecules24213916.

Many sentences are not understandable. Please overhaul the manuscript.

We have significantly revised MS.

Authors are not aware of recent publications related to this research direction. Some latest literature might have been missed; please make sure to update your references. 

Our reference list contains around 70% of 2019-2021 publications. We have added the latest relevant found also.   

Please revise your objective statement.

We have tried to do it.

Please mention the novelty and hypothesis as well at the end of the introduction.

We have commented in the text all major new points namely: i) an increase in CD13 and C90 expression by MSCs during osteogenic differentiation in the presence of all three types of NPs; ii) MSCs secretome was studied in dynamics of tri-lineage differentiation was done for the first time; iii) we have found the difference in the secretome between osteogenic and adipogenic differentiation; iv) we have shown the superior effect of AuNPs on the secretome. 

Results and discussion

What’s new finding in your synthesis methods? How your findings are novel and cost-effective as compared to previous reports.

We have used standard method of NPs synthesis. Finding are novel however cost-effectiveness can not be applied directly to our data. 

Why did you choose 25 mg/mL exposure dose? Please address your selection criteria?

This was the highest nontoxic dose, we have explained it in the text.

The discussion is less coherent and comprehensive, the authors need to look at the various aspect under different treatment conditions as a panorama and provide deeper discussion on the role of NMs effects.

We have revised the discussion.

and fig 4 (C-d) need to change the color scheme as current graphs not visible and understandable.

We have changed colors in Fig 4 (c-d) and was forced to change the colors in all other related figures.

Add a diagram illustrating a general mechanism behind your results.

We have no idea what should be shown in such diagram, at this stage of research we need more knowledge on the process.

During co-exposure of NMs treatment, have you performed any controlled experiment?

All the controls were included.

Please compare the toxic effects with other NMs and add the in-depth discussion. 1007/s11356-020-09565-8

NPs used in our work were nontoxic in the selected concentration

Conclusion

Conclusion is just a repetition of results. Please significantly improve this part.

We have tried to improve it, omitting some repetitions.

What’s practical application of your study? What’s new addition of your current experiment in terms of knowledge?

Clinical application of MSC for tissue regeneration is important. Any improvements can be of use. Knowing the mechanisms of NPs effects possibly will help to produce MSC of better quality.

Please add the mechanist illustration of your findings at the end.

We do not know these mechanisms, we are only stuffing them.

Round 2

Reviewer 1 Report

The authors have revised the manuscript a lot, but I still insist on using survival rate as the y-axis in the MTT results.

Author Response

Dear Ms. Roselyne Gao,

We were asked by Reviewer 1 to change Fig 2, d, however we have done it in our previous revision. We have added Y axis "Proliferation, OD, 540 nm" to make it more evident.

Academic Editor Notes: The authors are urged to include the values of the control in all of their experiments.

This is not clear to us: each experiment contain various controls (undifferentiated MSCs, treated with NPs only, treated with differentiation stimuli only, treated with both differentiation stmuli and NPs).